

# Walking and hypertension: greater reductions in subjects with higher baseline systolic blood pressure following six months of guided walking

Simona Mandini[1,2], Francesco Conconi[1], Elisa Mori[1], Jonathan Myers[3,4], Giovanni Grazzi[1,2] and Gianni Mazzoni[1,2]

[1] Center of Biomedical Studies Applied to Sport, University of Ferrara, Ferrara, Italy
[2] Public Health Department, AUSL Ferrara, Ferrara, Italy
[3] Health Care System Cardiology, VA Palo Alto Health Care System Cardiology, Palo Alto, CA, USA
[4] School of Medicine, Stanford University School of Medicine, Stanford, CA, United States of America

## ABSTRACT

**Background**. The aim of the study was to assess the effects of walking on the blood pressure in sedentary adults with differing degrees of systolic blood pressure (SBP).

**Methods**. A total of 529 subjects with SBP above 120 mmHg were enrolled. Blood pressure, body weight, body mass index, waist circumference and walking speed were determined at enrolment and after six months. Walking sessions were supervised by exercise physiologists.

**Results**. The weekly walking time of the subjects completing the project was uniform and reached 300 minutes by the second month. 56% of participants completed the 6 months intervention (182 women 59.6 ± 9.0 years, and 114 men, 65.4 ± 8.6 years) 27 had a baseline SBP >160 mm Hg, 35 between 150–159, 70 between 140–149, 89 between 130–139 and 75 between 120–129 mmHg. Following six months of supervised walking, SBP was significantly reduced in all subgroups ($p < 0.001$), with the greatest reduction (−21.3 mmHg) occurring in subjects with baseline SBP >160 and the smallest reduction (−2.6 mmHg) occurring in subjects with baseline SBP of 120–129 mmHg. Diastolic blood pressure, body weight, body mass index and waist circumference were also significantly reduced following the walking intervention ($p < 0.001$). These reductions were nearly identical within the various groups.

**Discussion**. In a large group of sedentary adults with varying degrees of SBP, 6 months of supervised walking elicited a marked reduction in systolic blood pressure with the largest reductions in pressure occurring in individuals with higher baseline SBP.

## INTRODUCTION

Hypertension is the most common, costly, and preventable cardiovascular disease risk factor (*Pescatello et al., 2015*) and is a major public health concern worldwide requiring intensive prevention and treatment management programs (*Ettehad et al., 2016*; *Redon et al., 2016*; *Whelton et al., 2017*). Regular physical activity reduces blood pressure and has

Corresponding author
Simona Mandini,
simona.mandini@unife.it

been recommended by American and the European hypertension guidelines (*Lemogoum et al., 2003*; *Corrao et al., 2011*). Recent meta-analyses have reported significant reductions in systolic and diastolic blood pressure among subjects who followed programs of regular walking (*Murtagh et al., 2015*; *Börjesson et al., 2016*). However, previous studies on the effects of aerobic exercise on blood pressure have in most cases included a mix of normotensive and hypertensive subjects (*Murtagh et al., 2015*; *Börjesson et al., 2016*). This has confounded the ability to analyze in detail the effect of physical activity on hypertension, given that blood pressure reductions appear to be more pronounced in subjects with more severe hypertension (*Murtagh et al., 2015*; *Börjesson et al., 2016*).

The 2017 AHA guidelines (*Whelton et al., 2017*) state that hypertension begins at systolic blood pressure (SBP) >130 mmHg and SBP of 120–129 mmHg is now considered elevated blood pressure.

Taking into consideration these guidelines (*Whelton et al., 2017*) and the significant reductions in systolic and diastolic blood pressure among subjects who followed programs of regular walking (*Murtagh et al., 2015*; *Börjesson et al., 2016*) our study has considered the effects of six months of guided walking on the blood pressure of 5 groups of subjects with baseline systolic pressure respectively between 120 and 130 mmHg, 130 and 140, 140 and 150, 150 and 160 and above 160 mmHg.

## MATERIALS & METHODS

### Subject recruitment

The study was advertised through local newspapers and emails sent to the employees of public organizations active in Ferrara, Italy. Adults and elderly subjects who declared a sedentary lifestyle were considered for the study. During the enrolment phase, face-to-face interviews with potential participants were conducted, during which the purpose and procedures of the study were explained. The Human Studies Committee of the University of Ferrara, number 22-13, approved the study. After obtaining an institutionally approved informed consent, in accordance with the Helsinki declaration, 529 subjects (327 women, 202 men) were enrolled over a 1-year period.

None of the patients enrolled presented with comorbidities that interfered with the exercise program of this study.

### Subject evaluation

Arterial blood pressure was determined in the seated position after three minutes of rest, using a validated automatic sphygmomanometer (Omron M3) and averaging the values of three successive determinations taken with two-minute intervals between each assessment. The instrument was calibrated each week. No physical activity was performed <3 hours prior to this evaluation. Subjects with a systolic blood pressure (SBP) >140 mmHg were admitted into the study upon authorization of their personal physician. Hypertensive therapies prescribed by family physicians were maintained and not modified during the intervention period. Height and weight were measured and body mass index (BMI) calculated accordingly. Waist circumference was measured using standard procedures (*American College of Sports Medicine, 2004*).

Walking speed was derived from the time taken to walk 100 m on a flat course (*Morice & Smithies, 1984*; *Rana, 2016*), at an intensity of 12–14 on the 6–20 Borg perceived exertion scale (*Borg, 1982*), with five minutes of slow walking preceding the test (*Grazzi et al., 2017*). The Borg scale was used to guide the subjects during the test for measuring their walking speed. An image indicating scales of work intensity with numbers (6 to 20), colors and images was presented and explained to the participants and shown before, during and after the test.

## Walking program

Initially, subjects began with a minimum of 15 to 30 minutes of daily walking and were encouraged to walk as frequently as possible with one of the walking groups using individualized walking speed organized within the project. The walking groups were guided and supervised by exercise physiologists. Walking sessions were performed outdoors, always on flat ground.

Walking groups were active twice a day, from Monday through Friday, in addition to one walking session on Saturday morning. Based on the walking speed measured at enrolment, the participants were initially assigned to walking groups of "slow" (up to 4 km/h), "medium" (4–5 km/h) or "fast" speeds (above 5 km/h). The participants walked five to six days a week, mainly within the walking groups. Walking speed and walking time were increased progressively. The increments of walking speed and duration were not forced but chosen spontaneously by the participants. The daily walking duration for each group increased to 50–70 minutes within two months and was maintained for the following four months. The weekly walking time was uniform and reached 300 minutes by the second month.

## Subject counselling

To motivate subjects and enhance compliance, a booklet on the importance of regular physical activity was distributed with suggestions on walking duration and intensity. To increase the sense of belonging to the project the participants were invited to monthly meetings on topics regarding the advantages of habitual physical activity and a healthy lifestyle.

## Post intervention assessments

Systolic and diastolic blood pressure, height, weight, BMI, waist circumference and walking speed were re-determined after six months.

## Statistical analysis

The assessed variables (systolic blood pressure, diastolic blood pressure, weight, BMI, waist circumference and walking speed) are expressed as mean ± standard deviation.

The Kolmogorov–Smirnov test was used to assess the normal distribution of all data.

Gaussian Curves were calculated to express the distribution of systolic pressure values at enrolment and after six months of walking in seven groups stratified by of different levels of baseline SBP. Differences between values measured before and at the end of the program were analyzed using paired samples Student's t-tests. ANOVA analysis were used

**Table 1  Systolic and diastolic blood pressure at baseline and after six months in five groups stratified by initial systolic blood pressure.**

| | n | Age (years) | Systolic blood pressure (mmHg) | | | | Diastolic blood pressure (mmHg) | | | |
|---|---|---|---|---|---|---|---|---|---|---|
| | | | Baseline | Six months | p | Δ | Baseline | Six months | p | Δ |
| Group 1 (>160 mmHg) | 27 | 66.2 ± 8.8 | 164.7 ± 7.5 | 143.3 ± 8.5 | *** | −21.3 ± 9.4 | 88.4 ± 9.9 | 81.1 ± 4.9 | ** | −7.3 ± 8.3 |
| Group 2 (150–159 mmHg) | 35 | 67.4 ± 5.6 | 152.1 ± 2.9 | 140.3 ± 6.4 | ** | −11.8 ± 6.9 | 81.9 ± 8.5 | 79.0 ± 6.6 | ** | −2.9 ± 7.3 |
| Group 3 (140–149 mmHg) | 70 | 64.3 ± 8.5 | 141.9 ± 2.5 | 134.4 ± 6.4 | *** | −7.5 ± 6.2 | 78.5 ± 8.8 | 76.6 ± 5.2 | * | −1.9 ± 7.7 |
| Group 4 (130–139 mmHg) | 89 | 61.2 ± 8.1 | 131.9 ± 2.5 | 126.6 ± 5.0 | *** | −5.3 ± 5.4 | 76.7 ± 7.8 | 74.5 ± 6.4 | ** | −2.2 ± 8.7 |
| Group 5 (120–129 mmHg) | 75 | 59.3 ± 9.7 | 122.7 ± 2.9 | 120.1 ± 4.3 | *** | −2.6 ± 4.3 | 75.1 ± 6.5 | 72.7 ± 5.7 | ** | −2.4 ± 5.8 |

**Notes.**
*$P < 0.05$.
**$P < 0.001$.
***$P < 0.0001$.

to compare the Δ change of the systolic blood pressure between sub-groups and to verify differences in body weight at baseline between each of the five group considered. The Bonferroni method was used for *post hoc* tests. The influence of age, weight, BMI, waist circumference and walking speed on SBP was analyzed by multivariate analysis. Differences were considered to be significant at the $P < 0.05$ level.

## RESULTS

A total of 296 (182 women 59.6 ± 9.0 years, and 114 men, 65.4 ± 8.6 years) subjects with baseline SBP above 120 mmHg completed the six month walking program.

No one experienced an adverse event during the study. The weekly walking time of the subjects completing the project was uniform and reached 300 minutes by the second month.

There were 119 subjects taking antihypertensive drugs; fifty-eight had a baseline SBP >140 mmHg; thirty had an SBP value between 130–139 mmHg; fifteen had an SBP between 120–129 mmHg. Anti-hypertension therapies were not modified during the study period.

There were 233 dropouts (168 women, 56.7 ± 11.7 years and 65 men, 65.3 ± 9.8 years). Most left the project within three months after enrolment for insufficient motivation, difficulties in scheduling, change of address and for other unknown reasons.

### Blood pressure changes relative to baseline systolic blood pressure

To analyze specifically the effects of walking on the blood pressure of the 296 subjects with elevated SBP finishing the project they were subdivided into five subgroups with baseline SBP >160 mmHg, between 150–159, between 140–149, between 130–139 and between 120–129 mmHg.

The values of systolic and diastolic blood pressure obtained after six month of walking in each of these subgroups are shown in Table 1.

SBP decreased significantly in each of the five subgroups, with lowering of $-21.3$ mmHg in the group of subjects with baseline values >160, $-11.8$ in the group 150–159, $-7.5$ in the group 140–149, $-5.3$ in the group 130–139 and $-2.6$ in the group 120–129 mmHg.

These reductions are also evident when analyzing the Gaussian curves of the SBP at baseline and after six months of walking in the five subgroups considered (Fig. 1). The relationship between decrease in SBP and the corresponding baseline value (Fig. 2) is described by a polynomial equation with $R^2$ of 0.98. This relationship was similar and equally significant when the data obtained in males ($R^2 = 0.97$) and in females ($R^2 = 0.99$) were analyzed separately (data not shown).

Diastolic blood pressure was also significantly reduced following six months of walking in all subgroups (Table 1). The decrease is more pronounced in the subjects with systolic pressure >160 mmHg and relatively uniform in the other four subgroups. The variations of $\Delta$ SBP between each of the five subgroups and their statistical significance are reported in Table 2.

### Anthropometric variables and walking speed modifications in the five subgroups with different baseline systolic blood pressure

Table 3 shows body weight, body mass index and waist circumference in the five subgroups considered at baseline and after six months. At baseline these variables were higher in the subjects of the subgroups with elevated SBP and progressively lower in the subgroups with lower SBP. No statistical differences was seen for the anthropometric variables considered within the five subgroups except for body weight which was higher in subgroup 3 relative to subgroup 4 ($p = 0.006$); in waist circumference which was higher in subgroup 1 relative subgroup 5 ($p = 0.01$) and for BMI which was higher in subgroup 2 and 3 relative subgroup 5 ($p = 0.04$). Following 6 months of supervised walking, body weight, body mass index and waist circumference decreased significantly in all subgroups. The reductions of the values of these anthropometric variables are relatively uniform and not correlated with the decrements of SBP (Table 3). Finally, walking speed increased significantly in all subgroups (Table 4).

## DISCUSSION

The main finding of the current study is that 6 months of supervised walking elicits significant reduction in SBP in a large group of sedentary adults with varying degrees of SBP. The six months walking program requiring a minimum of five workouts/week was rather demanding. This in part justifies the large number of dropouts.

The six months walking program completed by the sedentary subjects enrolled in the study has been followed by highly significant reductions of body weight, BMI and waist circumference.

These reductions, observed in the vast majority of the participants, are attributable to the extra calories expenditures requested by the walking program. Similar results were obtained in several studies carried out in sedentary non-hypertensive subjects (*Murtagh et al., 2015*; *Börjesson et al., 2016*). The increase in walking speed, observed in almost all finishers, indicates favorable functional and cardiovascular adaptations following the six

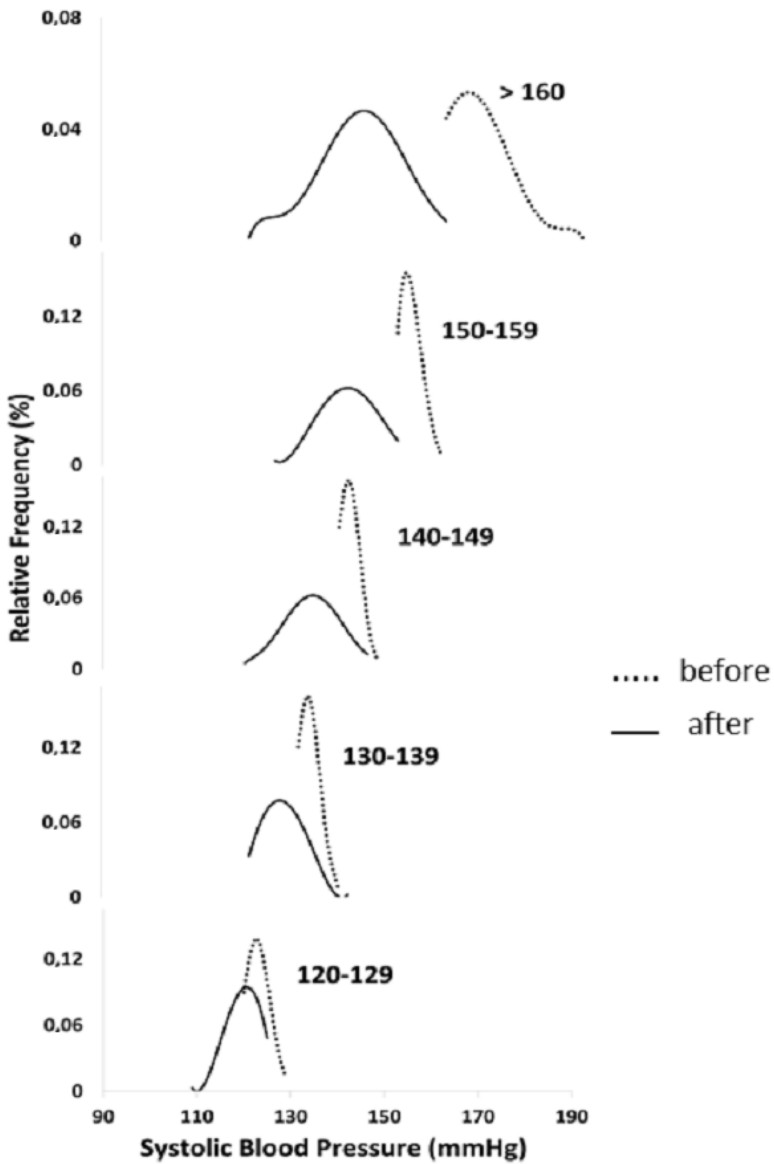

**Figure 1  Systolic blood pressure distribution at baseline and after six months of walking in five groups stratified by initial systolic blood pressure.**

months of almost daily physical activity. It is well noted that the reduction of body weight, BMI and waist circumference and the improvement of walking speed driven by regular physical activity are associated with reduction in cardiovascular risk (*Ross et al., 2016*). This is also shown in our study in which the other relevant cardiovascular risk factor, hypertension, is reduced.

One new finding is that the anti- hypertension effect is more pronounced in the subjects with baseline-elevated blood pressure. The SBP reduction varied from $-21.3 \pm 9.4$ in the group with SBP >160 mmHg to $-2.6 \pm 4.3$ in the subgroup with SBP between 120–129 mmHg. This observation is also evident from the Gaussian curves comparing the baseline

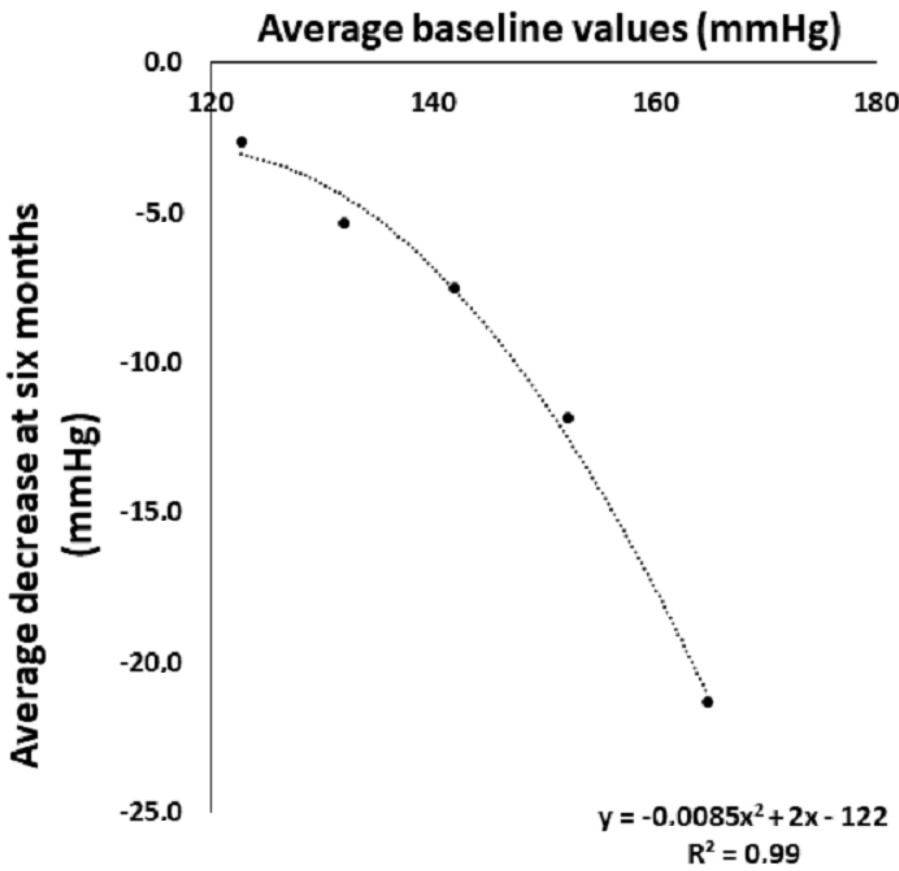

**Figure 2** Systolic blood pressure at baseline and its decrease after six months of walking in five groups stratified by initial systolic blood pressure.

SBP before and after 6 months in the five subgroups considered. The highly significant polynomial equation describing the relationship between SBP at baseline and its decrease after six months further emphasizes that in hypertensive sedentary subjects the greater the baseline SBP the greater its decrease after six months of walking. The reasons why the blood pressure lowering effect is much greater in the higher SBP subgroups is not explained by different body weight, waist circumference and walking speed since these values at baseline are uniform in the subgroups considered.

The blood pressure lowering effect of the walking program is not related to the decreases in body weight, body mass index or waist circumference, which are uniformly reduced between the subgroups with varying degrees of SBP. The improvements in walking speed (+0.6 km/h) are also evenly distributed among the subgroups and are not significantly related to decreases in SBP. In addition, the changes in SBP cannot be attributed to a therapeutic intervention since, during the six-month project, blood pressure therapy was not modified.

As documented by *Palatini et al. (2011)*, incremental increases in SBP generally occur progressively with aging and the widely used cut-off of 140 mmHg is often exceeded, leading

**Table 2  Comparisons of the decrease in systolic blood pressure (SBP) at six months between groups.**

| Δ SBP (mmHg) | Group comparison of the difference in decrease of SBP | | | | Standard Error | p | 95% CI |
|---|---|---|---|---|---|---|---|
| Group 1 | Group 1 | Vs | Group 2 | −9.926 | 2.400 | 0.0033 | −17.287 to −2.565 |
| −21.3 ± 9.4 | | Vs | Group 3 | −14.222 | 1.977 | <0.0001 | −20.284 to −8.160 |
| | | Vs | Group 4 | −14.222 | 2.370 | <0.0001 | −21.490 to −6.954 |
| | | Vs | Group 5 | −17.704 | 1.966 | <0.0001 | −23.734 to −11.674 |
| Group 2 | Group 2 | Vs | Group 3 | −4.296 | 1.646 | 0.1481 | −9.344 to 0.751 |
| −11.8 ± 6.9 | | Vs | Group 4 | −4.296 | 1.645 | 0.1477 | −9.341 to 0.749 |
| | | Vs | Group 5 | −7.778 | 1.802 | 0.0020 | −13.306 to −2.250 |
| Group 3 | Group 3 | Vs | Group 4 | 0.000 | 1.290 | 1.0000 | −3.956 to 3.956 |
| −7.5 ± 6.2 | | Vs | Group 5 | −3.481 | 1.358 | 0.1649 | −7.647 to 0.684 |
| Group 4 | Group 4 | Vs | Group 5 | −3.481 | 1.528 | 0.3114 | −8.167 to 1.204 |
| −5.3 ± 5.4 | | | | | | | |
| Group 5 | | | | | | | |
| −2.6 ± 4.3 | | | | | | | |

**Table 3  Average values for Body weight, Body Mass Index and Waist circumference at baseline and after six months in the five groups stratified by initial systolic blood pressure.**

| | Body weight (kg) | | | | Body mass index (kg/m$^2$) | | | | Waist circumference (cm) | | | |
|---|---|---|---|---|---|---|---|---|---|---|---|---|
| | Baseline | Six months | p | Δ | Baseline | Six months | p | Δ | Baseline | Six months | p | Δ |
| Group 1 (>160 mmHg) | 83.3 | 80.0 | *** | −3.3 | 28.8 | 27.7 | *** | −1.1 | 103.3 | 99.7 | *** | −3.6 |
| Group 2 (150–159 mmHg) | 82.9 | 80.1 | ** | −2.8 | 29.3 | 28.4 | ** | −0.9 | 103.0 | 99.8 | *** | −3.3 |
| Group 3 (140–149 mmHg) | 77.8 | 75.8 | *** | −2.0 | 27.6 | 26.9 | *** | −0.7 | 97.7 | 95.8 | *** | −1.9 |
| Group 4 (130–139 mmHg) | 74.1 | 71.7 | *** | −2.3 | 27.5 | 26.7 | *** | −0.9 | 96.0 | 92.4 | *** | −3.5 |
| Group 5 (120–129 mmHg) | 69.4 | 67.8 | *** | −1.6 | 25.4 | 24.8 | *** | −0.6 | 90.3 | 88 | *** | −2.3 |

Notes.
[**] $P < 0.001$.
[***] $P < 0.0001$.

to a high prevalence of hypertension in older subjects. In our hypertensive cohort of >60 years old individuals, 6 months of guided walking was effective in counteracting these age-related increases in SBP. The walking-dependent decrease of SBP we observed may be explained by several possible mechanisms. Prolonged walking may reduce sympathetic activity, increase vagal tone, or both, leading to a reduction in peripheral resistance (*Börjesson et al., 2016*). Regular physical activity may reduce norepinephrine levels by about 30% (*Fagard & Cornelissen, 2007*), and this reduction may parallel reductions in blood pressure (*Mancia et al., 2013*; *Duncan et al., 1985*). Another blood pressure lowering effect of physical activity is the release of vasodilating substances such as endorphins (*Thorén et al., 1990*) and reduced insulin resistance secondary to physical activity may

**Table 4** Average values for walking speed at baseline and after six months in the five groups stratified by initial systolic blood pressure.

| | Walking speed (km/h) | | | |
| --- | --- | --- | --- | --- |
| | Baseline | 6 months | p | Δ |
| Group 1 (>160 mmHg) | 5.6 ± 0.8 | 6.4 ± 0.9 | *** | 0.8 |
| Group 2 (150–159 mmHg) | 5.4 ± 0.7 | 6.0 ± 0.7 | *** | 0.6 |
| Group 3 (140–149 mmHg) | 5.6 ± 0.7 | 6.3 ± 0.6 | *** | 0.7 |
| Group 4 (130–139 mmHg) | 5.6 ± 0.7 | 6.4 ± 0.7 | *** | 0.8 |
| Group 5 (120–129 mmHg) | 5.7 ± 0.7 | 6.4 ± 0.5 | *** | 0.7 |

**Notes.**
*** $P < 0.0001$.

also play a role (*Rinder et al., 2004*). The blood pressure lowering effect could also be mediated by effects on kidney function (*Kenney & Zambraski, 1984*) through the reduction of plasma-renin levels (*Mancia et al., 2013*). Finally, a slight reduction in blood pressure could also be achieved through the effect of exercise on other risk factors, such as body weight and waist circumference (*Frisoli et al., 2011*; *Horvath et al., 2008*).

Habitual walking can therefore safely and effectively contribute to the blood pressure lowering in hypertensive subjects without exposing the patients to potential adverse effects of drug therapy.

Given the fact that moderate physical activity such as walking effectively lowers blood pressure and is associated with numerous other health benefits, guided walking programs should be included as standard adjunctive therapy for hypertension. Since walking groups are effective and safe, with a good adherence and wide-ranging health benefits, they should adopted as part of public health policy (*Hanson & Jones, 2015*).

### Limitation of the study

Previous randomized control studies have shown clearly that in control inactive subjects the variables considered, including systolic blood pressure, are not modified (*Murtagh et al., 2015*; *Börjesson et al., 2016*). For this reason, we chose to not include a control group and instead opted to include all subjects in the walking intervention.

## CONCLUSIONS

Six months of supervised walking in sedentary adults with high blood pressure is effective in reducing both systolic and diastolic blood pressures, with effects particularly evident in those with more severe hypertension.

### Abbreviations

| | |
| --- | --- |
| **MI** | Body Mass Index |
| **WC** | Waist Circumference |
| **SBP** | Systolic Blood Pressure |
| **WS** | Walking Speed |
| Δ | difference between blood pressure at baseline and after six months of walking |

### Funding

The study was supported by the Italian Ministry of Education and Scientific Research and the Ministry of Sport. The funders had no role in study design, data collection and analysis, decision to publish, or preparation of the manuscript.

### Grant Disclosures

The following grant information was disclosed by the authors:
Italian Ministry of Education and Scientific Research.
Ministry of Sport.

### Competing Interests

The authors declare there are no competing interests.

### Author Contributions

- Simona Mandini and Francesco Conconi conceived and designed the experiments, performed the experiments, analyzed the data, prepared figures and/or tables, authored or reviewed drafts of the paper, approved the final draft.
- Elisa Mori performed the experiments, analyzed the data, prepared figures and/or tables, approved the final draft.
- Jonathan Myers and Giovanni Grazzi approved the final draft.
- Gianni Mazzoni authored or reviewed drafts of the paper, approved the final draft.

### Human Ethics

The following information was supplied relating to ethical approvals (i.e., approving body and any reference numbers):

The Human Studies Committee of the University of Ferrara, number 22-13, approved the study.

### Data Availability

The database is included as Supplemental Information 1.

### Supplemental Information

Supplemental information for this article can be found online at http://dx.doi.org/10.7717/peerj.5471#supplemental-information.

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
