# Peer review of "Walking and hypertension: greater reductions in subjects with higher baseline systolic blood pressure following six months of guided walking"

_PeerJ, doi:10.7717/peerj.5471_

## Round 0.1 · original submission · Major Revisions

· Academic Editor

Major Revisions

The three reviewers have provided excellent feedbacks and comments for you to improve this work. We look forward to receiving your revised work in the near future.

·

Basic reporting

Clear and concise reporting throughout. The manuscript is generally well-referenced and structured appropriately. The figures and tables are presented professionally. The standard of English is good and suitable for publication.

Experimental design

The authors have presented important 'real-world' data regarding the efficacy of walking activity on resting BP in hypertensive patients, stratified according to baseline BP.

The methods are well-described and the research questions has been sufficiently outlined.
The experimental design has several important limitations. One suggestion might be as follows:
1. This is an uncontrolled study, absent of any control group not participating in a walking intervention. It would be interesting to compare these findings between participants who achieved a certain threshold of weekly walking versus those who remained relatively sedentary. Although this analysis would contain several biases, I believe it would strengthen the findings of this study.

Validity of the findings

I encourage the findings of this study, but we should be carful not to push these findings too far. This study has some important aspects that should be addressed, that will ultimately strengthen the conclusions.
1. Hypertensive patients are rarely free of comorbidities that may alter their participation or benefits from walking. Preferably, the authors should clarify these.
2. The authors suggest that medical therapy remained unchanged through the study, yet baseline medications are not reported in detail. It would be an advantage to include these perhaps in table format for clarity. Presumably, those patients with higher baseline BP may be those on more aggressive therapy (i.e. multiple agents)? The authors should consider whether a suitable adjustment can be made for this?
3. Please include mean and SD walking duration across all participants, rather than just stating that they walked up to 300 minutes per week (i.e. line 105+106), Did the mean walking duration differ according to baseline BP group?
4. Please try and avoid over-dichotomising. For example, figure 2 would benefit from including all patient data rather than just a group mean. This provides much more valuable information on the response according to baseline BP (including 95% CI of the regression to quantify variance).

Additional comments

The authors should be commended on an important analysis. I am confident that with some relatively minor changes, this will be an important study.

·

Basic reporting

Yes. Well-structured article. English is fine but I have made some recommendations to restructure sentences or rephrase things in places. Please see my minor revision section.

Experimental design

Please add following declaration of Helsinki when talking about informed consent process. Also please see my comments in major revisions section regarding additional analyses for comparing subgroups based on SBP.

Validity of the findings

I think that this work is impactful and novel in that it shows the importance of baseline systolic blood pressure on improvement. All hypertensive patients should be encouraged to exercise to achieve BP reductions but patients with SBP >150 mmHg in particular should incorporate moderate-intensity PA into their daily life as they see the biggest benefit. Statistics are fine but I would like to see additional analyses once the authors split into subgroups to ascertain whether changes are different by subgroup.

Additional comments

Major comments
1. I like how the statistics are done to compare pre-post changes in each subgroup but I would like to see additional analyses done to compare each of these subgroups. Should be pretty easy to do. You could run an ANOVA to compare the delta change between sub-groups. Right now you are saying the greatest improvement occurs in the higher baseline SBP group but you haven’t actually run stats to show that improvements in these groups were significantly greater than other sub-groups. Only that the value is higher.
Line 48: Please add that study procedures were carried out after obtaining written informed consent in accordance with the Helsinki Declaration. This is required by the journal.
Line 60: Did you control when and dosage of hypertensive drugs taken prior to each measurement? Just need to be consistent for example if patients took them a few hours prior to measurement at baseline needs to be the same or at least similar when you are re-evaluating them at 6 months.
Discussion
Line 142-147: Are these changes in body weight, BMI and waist circumference similar to other studies that have used walking interventions? Please discuss other studies and also how changes in these variables may reduce CVD risk.
Line 146-147: Are increases in walking speed associated with reductions in CVD risk? Improvements in fitness and functional capacity? Tons of work by Blair, Myers, etc on this topic I believe please cite some of this work here.
Line 157-159: BP lowering is much larger in the higher SBP groups yet body weight is reduced uniformly across subgroups therefore BP lowering is not entirely driven by changes in body weight as you correctly point out. Why do you think the improvement is so much greater in the higher SBP subgroups? Feel free to speculate with a few sentences and use the literature to guide you here. Differences in baseline PA levels in addition to higher SBP maybe?
Line 197-201: Unless this is part of the journal instructions, I recommend moving this limitations section to the last part of the discussion. So it will now be the last paragraph of the discussion.
Line 199-201: Please insert citations for these previous randomized controlled trials you mention in this sentence.
Conclusion:
I really like the conclusion. You took your findings and applied it clinically. Keep as is.
Minor comments/revisions
Abstract
Line 2: Change to the following… “The aim of the study was to assess the effects of walking on blood pressure in sedentary adults with differing degrees of hypertension.”
Line 6: Change to the following… “Walking sessions were supervised by an exercise physiologist.”
Line 7: Change to the following… “The average weekly walking time during the six month study was xxx ± xx. Please insert mean ± SD for number of minutes walked per week where you see xxx ± xx.
Line 7-17: You don’t need to start a new line at line 8 or line 11 in the results section of the abstract group all sentences together in the abstract results section.
Line 12: Change this sentence to start as “Following 6 months of supervised walking, systolic blood pressure reduced significantly in each subgroup with lower of…” Also please add a p-value (p<0.001 or p<0.05) at the end of the sentence in parentheses.
Line 15: Change this sentence to “Diastolic blood pressure, body weight, body mass index and waist circumference were also significantly reduced following the walking intervention.” Also please add a p-value (p<0.001 or p<0.05) at the end of the sentence in parentheses or after each variable.
Line 17: Remove this sentence “Walking speed increased significantly and uniformly in all groups.” This is an interesting finding and definitely belongs in the main body of the paper but it is not needed in the abstract, as your main variables of interest are blood pressure.
Line 18: Consider rephrasing to as follows: “In a large group of sedentary adults with varying degrees of hypertension, 6 months of supervised walking elicited a marked reduction in systolic blood pressure with the largest reductions in pressure occurring in individuals with higher baseline systolic pressure.”
Introduction
Line 38: Rephrase to read as follows “Taking into consideration the 2017…”
Materials and Methods
Line 46: I would change to “The study was advertised through local newspapers…”
Line 58: Change to “No physical activity was performed <3 hours prior to this evaluation.”
Line 89: Please add blood pressure here. It is your main outcome variable.
Line 100: Excellent job on stats very clean and easy to follow. See comment about additional analysis in the Major revisions section.
Results
Line 104: Change to “six month walking program”
Line 105: Did everyone do 300 minutes per week? Would be great to see Mean ± SD for minutes per week if you have that data.
Line 108: Thank for including this information about dosage of drug not changing during the intervention.
Table 1, Table 2 and Table 3: please add units of measure for the variables measured. For example body weight (kg), body mass index (kg/m2), etc…
Table 2 and Table 3: please add standard deviatons.
Line 127: Please consider rephrasing to “Diastolic blood pressure was also significantly reduced following six months of walking in all subgroups (Table 1).”
Line 128: See my comment about adding additional analysis to assess differences between subgroups.
Line 133: How do you know this? Are the statistically higher? Consider analyzing and simply adding to Table 2 if significant differences exist at baseline for anthropometric variables.
Line 135: Please consider rephrasing to “Following six months of supervised walking, body weight, BMI and waist circumference decreased significantly in all subgroups.”
Line 138: consider adding data not shown at the end of this sentence. I totally believe that there was no correlation but just add it in.
Line 138: consider rephrasing to “Finally, walking speed increased significantly in all subgroups”.
Discussion
Line 142: Move line 148 to here. Start your discussion with your main findings sentence. Rephrase it start the discussion with “The main finding of the current study is that six months of supervised walking elicits significant reductions in systolic blood pressure a large group of sedentary hypertensive adults with varying degrees of hypertension.”
Line 142-147: move this paragraph to be the 2nd paragraph of the discussion. Keep the 1st paragraph only about your main findings and blood pressure.

·

Basic reporting

In the main, the manuscript has clear professional English. The article is brief and to the point, which this reviewer appreciates. However, I will point out that the introduction (and discussion) requires further 'balancing' of literature. There is an extensive body of research in physical activity and blood pressure (particularly epidemiology) which should be drawn upon to allow narrowing of your focus to a hypothesis (at the end of your introduction).

Experimental design

The manuscript presents original research, with ethical clearance that falls withing the Scope of PeerJ. The research question is relevant but lacks a meaningful definition. For instance, there is no hypothesis to which the article can be held accountable to. This has important ramifications for methods/results/discussion. However, this can be fixed.

Technical aspects:
https://doi.org/10.1161/HYP.0000000000000066 Hypertension. 2018;71:1269-1324

blood pressure diagnosis, using the criteria above, uses an average of ≥2 readings obtained on ≥2 occasions to estimate the individual’s level of BP. At the first visit, record BP in both arms. Use the arm that gives the higher reading for subsequent readings and separate repeated measurements by 1–2 min.This does not appear to have been adhered to in the present manuscript. Further, patients/subjects should avoid caffeine, exercise, and smoking for at least 30 min before measurement. Investigators should use a BP measurement device that has been validated, and ensure that the device is calibrated periodically. Similarly, investigators should use the correct cuff size, such that the bladder encircles 80% of the arm, and note if a larger- or smaller-than-normal cuff size is used, thus ensuring that proper measurements needed for diagnosis and treatment of elevated BP/hypertension

Validity and reliability of the walking test is not presented. The measurement of blood pressure requires further clarity (see below for more detail). These aspects may seem trivial but the authors need sufficient detail as to allow the study to be replicated.

The narrative dialogue of the present exercise programme requires further details of consistency. There is no reporting of subjective / objective determination of activity level prior to enrollment to the study. This is normal for physical activity studies. The methods section It is lacking in detail normally expected. For instance, exercise AND counseling is not an exercise program per se, it is a combined program. It is critical for these points to b acknowledged.

Validity of the findings

Data do not match that reported within the study. Specifically, 520 recruited, 284 dropped out and analysis was performed on 296 ‘hypertensives’. This appears to be a miscalculation.

If we forget the miscalculation, there remains two important problems for the authors to resolve;

1. There is an inherent sample bias in selecting to analyze those that ‘survived’ a 6 month walking program. It would be of significant value if you had intermediary data in those that dropped out. This would allow you to employ a quasi ‘intention to treat' analysis.

2. Under the 2017 AHA (et al.) clinical guidelines specified in your introduction, ‘hypertension’ begins at SBP>130mmHg, whereas SBP:120-129mmHg is classified as ‘elevated blood pressure’. The authors should address this important nomenclature throughout the document. This becomes critical in this work because 'elevated blood pressure' requires the combination of SBP and DBP to be presented. This is of major importance to improve this article.

In my view, the statistical analysis of data requires some focus. I will leave this with the authors apart for indicating that group x time comparison of multiple data requires a multivariate analysis of variance. Similarly, where categorical analysis is being used (ie...movement from categories of hypertension), particular (non linear) statistics should be employed. That is, you should allow for multiple categorical comparisons.

It is difficult for me to proceed with interpretation of this study as a reanalysis that addresses the reporting bias (ie..those who dropped out) is important to make a meaningful interrogation of your discussion and conclusion. However, I would suggest that this should be done for any re-submission.

Additional comments

Dear Author team,

Please take these comments in the spirit that they are meant. In my opinion, I believe that there are important limitations that have not been acknowledged and some key methodological issues that you will need to address. Encouragingly, it seems that many of these can be fixed and in doing so, you will have a manuscript that provides some value to the scientific literature.

Best wishes.

---

## Round 0.2 · Minor Revisions

· Academic Editor

Minor Revisions

Your manuscript has been reviewed by 2 of the 3 original reviewers. Both have suggested minor revisions. In addition to their suggestions, I have the following comments for your considerations:

1. Title of manuscript: Please revise to "Walking and hypertension: greater reductions in subjects with higher systolic blood pressure after six months of guided walking"
2. Line 143-5: Please add "to" after the words "relative"
3. Line 165: "It well none" should read "It is well noted"
4. Line 188-189: This can be altered as follows: In our hypertensive cohort of >60 years old individuals, 6 months of guided walking was effective in counteracting these age-related increases in SBP.
5. Delete Line 204 in conclusions: "Walking is also associated with a reduction in diastolic blood pressure."
6. Change Line 202-203 to: In conclusion, our study shows that in sedentary adults with high blodo pressure, walking is effective in reducing both systolic and diastolic blood pressures, with effects particularly evident in those with more severe hypertension.
7. Lines 206-212: Should be moved out of "conclusions", please make this the last sentences before 'study limitations'. Further, "Since walking groups are effective and safe, with a good adherence and wide-ranging health benefits, they should become part of the health system" can be improved as follows: "Since walking groups are effective and safe with wide-ranging health benefits, they should be adopted as part of public health policy."
8. Table 2: Please change title to: "Comparisons of the decrease in systolic blood pressure (SBP) at six months between groups"

·

Basic reporting

Yes. Well-structured article. English is fine but I have made some recommendations to restructure sentences or rephrase things in places. Please see my minor revision section.

Experimental design

No comment.

Validity of the findings

I think that this work is impactful and novel in that it shows the importance of baseline systolic blood pressure on improvement. All hypertensive patients should be encouraged to exercise to achieve BP reductions but patients with SBP >150 mmHg in particular should incorporate moderate-intensity PA into their daily life as they see the biggest benefit.

All of my comments in follow-up review are to improve the readability of the published paper.

Additional comments

Great job on the responses and revisions. Please strongly consider making a few minor changes to the final version that will appear in print as I feel these will greatly enhance the readability of the paper.

·

Basic reporting

Sound.

Experimental design

Sound.
For clarity within the abstract, I recommend replacing "Of the 296 subjects completed the project" with "56% of participants completed the 6 month intervention".

Validity of the findings

Sound

Additional comments

The manuscript has been significantly improved and worthy of publication.

---

## Round 0.3 · accepted · Accept

· Academic Editor

Accept

I provide a few minor edits in the attached document. Many thanks for resubmitting so promptly.